# CLiMRS: Cooperative Large-Language-Model-Driven Heterogeneous Multi-Robot System

## Abstract

Cooperative multi-robot tasks often require heterogeneous agents to collaborate over long horizons while managing spatial constraints and execution uncertainties. Although large language models (LLMs) excel at reasoning and planning, their potential for coordinated control in heterogeneous multi-robot teams has not been fully explored. We present **CLiMRS**, an adaptive negotiation framework inspired by human teamwork. The framework pairs each robot with an independent LLM agent and dynamically forms subgroups to facilitate perception-driven discussions and collaborative planning under long-horizon uncertainty. Within each group, local oracle planners lead parallel discussions to synchronize actions, while agents provide feedback to refine plans. This grouping–planning–feedback–execution loop enables efficient long-horizon planning and robust execution. To evaluate these capabilities, we introduce **CLiMBench**, a heterogeneous multi-robot benchmark of challenging assembly tasks with diverse robot types and skill libraries. Across both **CLiMBench** and a simpler benchmark, **CLiMRS** surpasses the best baseline, boosting success rates and improving efficiency by over 40% on complex tasks while maintaining very high success on simpler tasks. Our results demonstrate that leveraging human-inspired group formation and negotiation principles markedly enhances the efficiency of heterogeneous multi-robot collaboration.

## 1 Introduction

Addressing real-world, everyday tasks often requires collaboration to enhance the efficiency of long-horizon, complex planning and perception. Meanwhile, the development of intelligent agents that can assist embodiments in accomplishing such tasks remains an open challenge, particularly regarding how these agents can effectively help humans and other robots execute such intricate operations. Inspired by human teamwork, incorporating principles of human teaming into multi-agent systems, where sub-groups coordinate planning and perception through shared observations and information, offers a promising yet challenging path to improving efficiency and robustness Zhang et al. (2024b).

At the same time, large language models (LLMs) have exhibited outstanding performance across various dimensions, including natural language question answering Rein et al. (2024), code generation Jain et al. (2024), and logical reasoning Plaat et al. (2024). In recent years, numerous studies have integrated LLMs into robotic planning scenarios Song et al. (2023); Zhang et al. (2024a); Mower et al. (2024); Salimpour et al. (2025); Liang et al. (2025), with some extending their application to multi-robot collaborative planning tasks Zhang et al. (2024b); Mandi et al. (2024); Liu et al. (2025).

However, earlier explorations of robot collaboration largely center on homogeneous agents, which restricts the range of capabilities that can be demonstrated Liu et al. (2024a). Furthermore, works on heterogeneous teams typically assume ideal operating conditions Liu et al. (2025), while such assumptions ignore the cumulative errors that escalate over long horizons, driving up communication costs and undermining cooperative efficiency. While these advances show the promise of LLM-driven multi-robot collaboration, important gaps persist when the setting involves heterogeneous agents, long-horizon objectives, and the practical constraints of real-world operation.

To address these limitations, we propose **CLiMRS** (**C**ooperative Large-**L**anguage-Model-Dr**i**ven Heterogeneous **M**ulti-**R**obot **S**ystem), a human-team-inspired LLM-driven adaptive-negotiation framework that orchestrates heterogeneous robots through dynamic sub-group formation and cooperative planning, supporting robust long-horizon collaboration in uncertain environments.

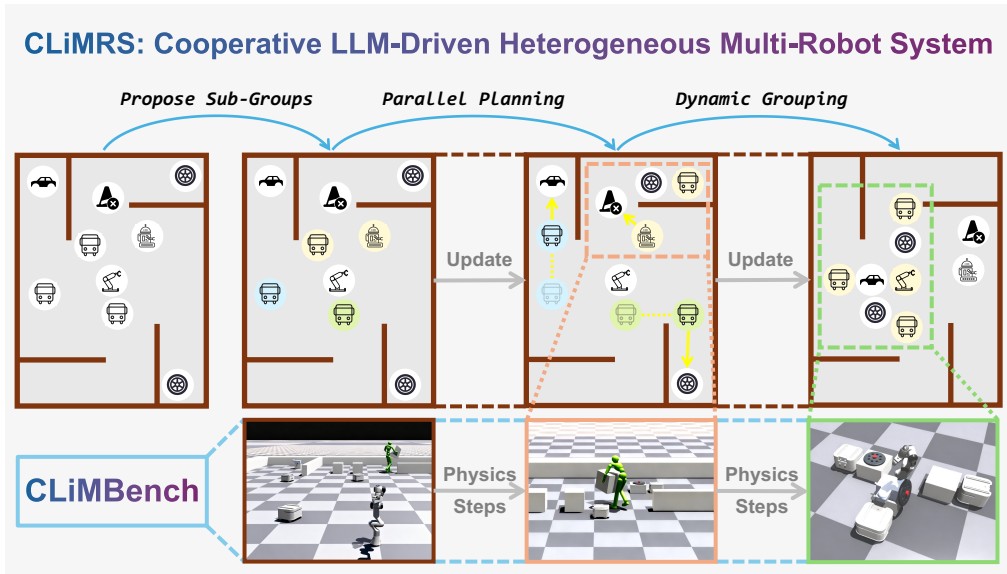

Figure 1: **Overview.** We present **CLiMRS**, a human-team-inspired negotiation paradigm for heterogeneous multi-robot systems that dynamically forms perception-driven discussion sub-groups, and **CLiMBench**, a heterogeneous multi-robot benchmark with challenging assembly tasks.

In this framework, each robot is guided by an independent LLM agent that communicates with peers to accomplish complex, long-horizon tasks. To strengthen collaborative effectiveness, the system leverages the broad world knowledge of LLMs and explicitly models inter-agent dependencies through a carefully designed grouping–planning–feedback–execution loop.

With **CLiMRS**, we further explore its applicability to challenging industrial scenarios, where heterogeneous robots must handle unpredictable execution errors. To evaluate this, we introduce **CLiMBench**, a benchmark for heterogeneous multi-robot collaboration. It features five robotic devices across three types of heterogeneous robots, equipped for transportation, conveyance, and assembly. Tasks of varying difficulty simulate material-handling and assembly processes with diverse skill usage, designed to test the planning and perception capabilities of LLM-based frameworks.

We evaluated our proposed framework in two distinct environments: **CLiMBench** and another heterogeneous robot collaboration benchmark Liu et al. (2025). Our experiments show that **CLiMRS** outperforms the best baseline, increasing success rates and improving efficiency by over 40% on complex tasks while maintaining high success on simpler ones. These results demonstrate that incorporating human-inspired group formation and negotiation principles substantially enhances the efficiency of heterogeneous multi-robot collaboration. To summarize, our main contributions are:

- We present **CLiMRS**, a multi-LLM cooperation framework for heterogeneous multi-robot collaboration which can perform long-horizon planning and efficient perception in complex tasks.
- We propose **CLiMBench**, a benchmark evaluating heterogeneous multi-robot collaboration in industrial assembly scenarios, featuring varied skill sets and a realistic simulation environment.
- We demonstrate through extensive experiments that **CLiMRS** achieves significant efficiency improvements via dynamic group formation and cooperative long-horizon planning.

## 2 RELATED WORK

### 2.1 EMBODIED SKILLS TRAINING ACROSS DIVERSE SCENARIOS

**Embodied Agent Skill Training.** Approaches to train embodied skills for task execution generally follow two primary paradigms: rule-based and learning-driven methods. Traditional embodiment controllers optimize joint movements through the resolution of robotic kinematics, aiming to im-

prove motion robustness and generate smoother, more precise trajectories Kashyap & Parhi (2021); Katayama et al. (2023). In recent years, with the advances of reinforcement learning and imitation learning in robotic motion control, spanning domains such as dexterous manipulation Rajeswaran et al. (2017); Zhu et al. (2019); Chen et al. (2022); Luo et al. (2025), bipedal locomotion Li et al. (2025); Zhang et al. (2024c); Serifi et al. (2024), and quadrupedal navigation Bellegarda et al. (2024); Shi et al. (2024), embodied perception has progressively learned to coordinate actions in a cerebellum-like fashion, enabling increasingly complex tasks in diverse environments. Overall, as tasks and environments grow more complex, embodied intelligence is shifting from traditional low-level planning toward more integrated, end-to-end perception and control.

**Multi-agent Skill Training.** Originally developed in game AI Kurach et al. (2020); Perolat et al. (2022), multi-agent skill training has since extended to industrial fields such as robotics Wang et al. (2024); Lai et al. (2025) and autonomous driving Li et al. (2022), where many of the coordination and credit-assignment strategies first pioneered in games remain fundamental. Despite these advances, current methodologies for multi-agent embodied tasks remain underdeveloped, particularly in light of the exponential state-space challenges introduced by an increasing number of robotic agents. Although certain researchers have explored mean-field approximations to alleviate these challenges Yang et al. (2018), robust generalization across heterogeneous robots has yet to be realized.

To further this goal, we design a set of generalizable robotic skills in **CLiMBench** to support heterogeneous multi-agent collaboration, leveraging robots' low-level control capabilities for high success rates and reducing the impact of execution failures on higher-level task planning.

## 2.2 TASK PLANNING WITH LLMS IN ROBOTICS

**LLM Planner for Robotics.** The rapid progress of LLMs in generalization and commonsense reasoning has fueled growing interest in robotics, as their strong few-shot Brown et al. (2020); Madaan et al. (2022) and zero-shot Huang et al. (2022); Kojima et al. (2022) learning capabilities make them well-suited as task planners for robots. Reliable code-generation abilities further allow LLMs to synthesize precise, executable instructions for robotic control Liang et al. (2023a); Singh et al. (2023); Wang et al. (2023); Wu et al. (2023a), and value-function-based approaches Lin et al. (2023); Ahn et al. (2022) leverage these models to select robust, skill-level commands for robotic agents. Recent improvements in context-driven prompting strategies Zhang et al. (2024b); Mandi et al. (2023); Liu et al. (2025); Wu et al. (2023b) have strengthened LLM-based task planning even further. Moreover, some studies Mandi et al. (2023) demonstrate that LLMs can reason and plan directly in 3D joint space, enabling the generation of fine-grained and precise task instructions.

**Multi-LLM Task Planning.** A promising way to overcome the limits of a single LLM in complex reasoning is to use multiple LLMs with cooperation, employing strategies such as round-table discussion Chen et al. (2023a), mutual debate Liang et al. (2023b), and role assignment Hong et al. (2024) to divide labor and improve output reliability. In embodied tasks, many studies emphasize the use of feedback Mandi et al. (2023); Liu et al. (2025) and memory modules Zhang et al. (2024b); Mandi et al. (2023); Liu et al. (2025); Wang et al. (2023) to enhance multi-LLM perception and planning. These modules allow LLMs to generate execution-level feedback and refine planning decisions using the rich context stored in well-designed memory components.

**Decision Paradigms in Multi-Robot Collaboration.** Two primary decision-making paradigms have emerged for complex multi-robot tasks: centralized and decentralized approaches. In decentralized schemes, multiple models or agents communicate, exchange intermediate plans, and iteratively refine their decisions through structured dialogue Mandi et al. (2023); Zhang et al. (2024b); Liu et al. (2024b),while centralized methods typically rely on a single, large-scale LLM to decompose global objectives and allocate tasks when planning Kannan et al. (2023); Liu et al. (2025). A recent comparative study conducted across four diverse multi-agent 2D scenarios Chen et al. (2023b) further reports that centralized communication consistently achieves higher success rates and markedly greater token efficiency, highlighting its strong potential for scalable real-world deployment.

To enhance collaboration in multi-robot scenarios, we propose a multi-LLM cooperation framework inspired by human teamwork. Robots are organized into dynamic subgroups for specific sub-tasks, reducing communication overhead while enabling concurrent discussions, plan refinement, and parallel action execution to improve efficiency and maintain a high success rate.

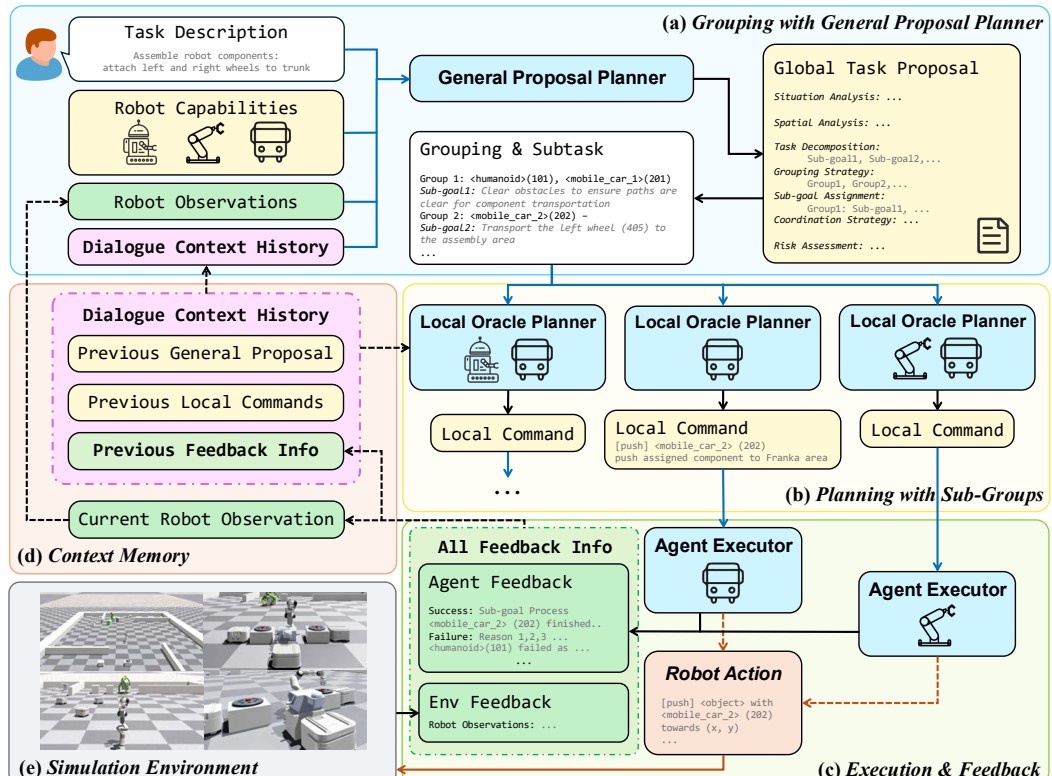

Figure 2: **CLiMRS Framework.** To employ our grouping–planning–feedback–execution cycle, **CLiMRS** comprises (a) a general grouping module, (b) multiple local planners, (c) multiple agent execution and feedback modules, (d) a context memory module, and (e) a simulation environment.

## 3 METHOD

In this section, we present **CLiMRS**, an adaptive negotiation-driven multi-LLMs cooperation framework for heterogeneous robot systems. Inspired by human teamwork, our approach forms dynamic agent sub-groups that facilitate centralized discussions on robot perception in parallel, with each robot paired with an individual LLM agent to give feedback to these discussions, resulting in a dynamic grouping–planning–feedback–execution cycle. As illustrated in Fig. 2, **CLiMRS** comprises five core modules: (a) a general grouping module that forms dynamic agent groups, (b) multiple local planners that generate agent commands, (c) agent execution and feedback modules that produce robot skills and return execution feedback, (d) a context memory module that records all inter-agent dialogues, and (e) a simulation environment for real-time interaction.

### 3.1 GROUPING WITH GENERAL PROPOSAL PLANNER

The first stage of our grouping–planning–feedback–execution cycle is to dynamically partition the agents into sub-groups, each responsible for different aspects of the overall task. To achieve this, we use a *general proposal planner* to augment the task instructions and orchestrate the grouping process.

**General Proposal Planner.** As illustrated in Fig. 2(a), the *general proposal planner* generates a global task proposal that organizes all agents into sub-task-oriented teams. Given the overall task instruction, this prompted LLM incorporates robot capabilities, current observations, and the dialogue history through a structured prompt. It outputs a well-defined plan designed to facilitate systematic reasoning: (1) *Situation Analysis*, assessing the environment and the current progress of the task; (2) *Spatial Analysis*, accounting for the locations of agents and known objects, as well as spatial constraints; (3) *Task Decomposition*, breaking the objective into executable sub-tasks; (4) *Grouping Strategy*, deciding how to cluster agents for concurrent or parallel work while minimizing interference;

(5) *Sub-goal Assignment*, specifying the objective of each group; (6) *Coordination Strategy*, outlining inter-group synchronization and execution order; and (7) *Risk Assessment*, identifying potential conflicts and corresponding mitigation plans. The resulting mapping from agent groups to their designated sub-tasks is then extracted and passed to the perception and execution modules.

## 3.2 PLANNING WITH SUB-GROUP LOCAL PLANNERS

Given the agent groupings and their designated sub-tasks, the second stage of our cycle issues precise commands to individual robots according to their capabilities and current observations. Because these sub-tasks are mutually independent, multiple *local oracle planners* operate in parallel to generate commands for different robots simultaneously, which is shown in Fig. 2(b).

**Local Oracle Planner.** The *local oracle planner* facilitates a centralized discussion among robots in a sub-group to determine precise commands for completing their assigned sub-tasks. This discussion leverages prior agent feedback stored in the dialogue context history. Similar to the *general proposal planner*, the *local oracle planner* takes into account sub-task instructions, robot capabilities, partial observations, and historical dialogue as context, but operates within a narrower scope to make fine-grained decisions focused on individual agents executing specific skills.

## 3.3 AGENT EXECUTION AND FEEDBACK

With commands issued to the robots, the final two stages of our cycle require them to evaluate these commands, determine appropriate actions, and provide feedback to refine future planning while ensuring safe execution. The *agent executor* LLM verifies the feasibility of its command and issues the corresponding action only when the command is deemed executable. The feedback then consolidates outcomes from both the LLMs and the simulator, gathering information to guide subsequent planning cycles and thereby closing the loop of negotiation among the LLMs.

**Agent Execution with Feedback.** Shown in Fig. 2(c), the *agent executors* verify and execute commands from *local oracle planner* while providing feedback. Each *agent executor* LLM considers its robot's capabilities, current observations, and available actions. The executor first checks its feasibility against the robot's physical constraints and observations. If feasible, the action is executed using the robot's skills; otherwise, the robot remains idle in this loop. Simultaneously, the executor produces feedback based on its evaluation, which is sent to the feedback module to inform future planning. Execution failures are categorized as (1) *improper grouping*: no robot in the group can complete the sub-task; (2) *incorrect agent selection*: a valid sub-task is assigned to an unsuitable robot; and (3) *state inconsistency*: missing information or unmet conditions prevent execution. For successful actions, the module also evaluates whether the sub-task has been fully accomplished.

**Feedback Formation.** The feedback is aggregated from two sources: (1) environmental observations updated after robot actions are executed in the simulator, and (2) outputs from the *agent executors*. This information is then integrated into the *context memory* for the next grouping–planning–feedback–execution cycle. The feedback both guides the *general proposal planner* during grouping (Sec. 3.1) and aids centralized discussions by the *local oracle planners* (Sec. 3.2). In this way, the accumulated observations and executor outputs provide essential context for refining both the global task proposal and the detailed local commands.

## 3.4 CONTEXT MEMORY AND ENVIRONMENT

Following the grouping–planning–feedback–execution cycle described above, our framework depends on two essential modules to make the workflow of the entire cycle operate smoothly: the *context memory* module and the *simulation environment*.

**Context Memory.** As shown in Fig. 2(d), the *context memory* collects (1) current feedback and planning dialogue together with the dialogue history from previous cycles, (2) robot observations from the *simulation environment*, and (3) the latest outputs from agents and planners. For the *general proposal planner*, it retains the previous five dialogue turns and the newest observations, allowing agent feedback to inform new proposals and groupings. For the *local oracle planners*, it stores each group's latest observations and the last five dialogue turns, providing rich situational context to guide and refine subsequent planning decisions.

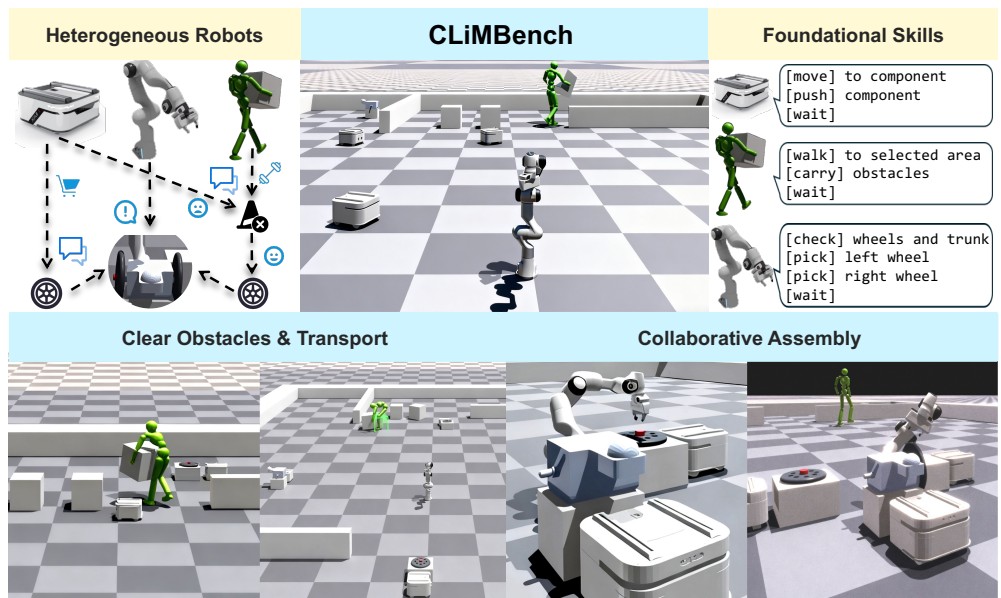

Figure 3: **CLiMBench Benchmark. CLiMBench** is a heterogeneous multi-robot collaboration benchmark designed to evaluate **CLiMRS**. It features multi-agent robots with diverse skills, enabling collaboration on tasks like transportation, conveyance, and assembly across varying difficulty levels.

**Simulation Environment.** Shown in Fig. 2(e), the simulation environment serves as the execution backbone of our framework. It receives the robot skill execution signals issued by the *agent executors* and immediately carries out the corresponding low-level actions in real time. During execution, it monitors the evolving state of the environment and produces both updated robot observations and environment-level feedback. These outputs are fed back to the *context memory*, allowing the overall method to track task progress, refine its understanding of the environment, and supply the information required for the next round of the grouping–planning–feedback–execution cycle.

## 4 BENCHMARK

In this section, we present **CLiMBench**, a benchmark for heterogeneous multi-robot collaboration. As shown in Fig. 3, we construct an assembly environment in IsaacGym Makoviychuk et al. (2021) that features diverse robotic agents and modular components. To enable effective integration with LLM-based planning, robot actions are executed by invoking predefined skills.

Unlike some other multi-agent collaboration benchmarks Liu et al. (2025), which decouple skill execution from the planning–execution loop and assume that all robot skills succeed by default, **CLiMBench** executes every robot skill within a realistic physics simulation, enabling genuine interaction between planning and execution. This distinction is critical because collaborative assembly tasks are inherently difficult, demanding not only high-precision manipulation but also effective coordination among multiple agents. The following subsections describe the scene construction and skill design mechanisms of **CLiMBench**, and additional details are provided in the Appendix.

### 4.1 SCENE CONSTRUCTION IN **CLiMBench**

**CLiMBench** features an industrial assembly scene that includes both assembly components and robotic agents. To increase task complexity and enhance realism, we introduce blocking obstacles into the environment settings. As is illustrated in Fig. 3, our robotic arm is implemented using a Franka Emika Panda arm, the AGV platform is based on the TRACER Mini robot, and the humanoid is implemented using the virtual humanoid agent.

**Scene Initialization and Randomization.** We initialize the environment and introduce controlled variations in task parameters and object configurations to enhance generalization. At the start of

Table 1: **Robot Skill List in CLiMBench.** We assign each robot type a distinct set of skills in **CLiMBench** based on its specific capabilities.

| Robot type | Num | Skill list |
|---|---|---|
| Robotic Arm | 1 | [check] <franka>check <trunk>
[check] <franka>check <left wheel>
[check] <franka>check <right wheel>
[pick] <franka>pick and place <left wheel>on <trunk>
[pick]  <franka>pick and place <right wheel>on <trunk>
[wait]  <franka>wait |
| AGV | 3 | [move] <mobile_car>move to component location using RRT path
[push] <mobile_car>push selected component to franka area
[wait] <mobile_car>wait |
| Humanoid | 1 | [walk] <humanoid>move to selected area
[carry] <humanoid>carry <obstacles>
[wait] <humanoid>wait |

each episode, robots execute their skills under randomized task conditions, leading to diverse skill sequences and varying levels of inter-agent synchronization. This setup provides a robust testbed for evaluating the effectiveness of different LLM architectures in multi-agent collaborative tasks.

**Environment feedback.** We design the environment feedback along two dimensions: (1) updating the state of all agents and the coordinates of objects within their perceptible range, and (2) reporting conflicts that arise when multiple robots execute skills simultaneously.

### 4.2 ROBOT SKILL DESIGN IN **CLiMBENCH**

In **CLiMBench**, each robot receives both the global task objectives and observations pertinent to its specific skill set (e.g., a humanoid robot observes its joint states, torso status, and target positions). This requirement makes it essential to clearly specify how each agent's designated skills are implemented in practice in **CLiMBench**. Summarized in Table 1, we design distinct skill sets for different types of robots, with some other details provided in the Appendix.

**Robotic Arm Manipulation with Franka.** We employ a two-stage control strategy to balance speed and precision. The Franka arm first executes a rapid coarse motion, then slows for fine adjustment to ensure accurate placement. An operational-space controller (OSC) uses the task-space inertia matrix and gravity compensation to compute joint torques, yielding a spring–damper response Narang et al. (2022). Smooth, continuous waypoints are generated by interpolation for reliable execution.

**AGV Transportation with TRACER Mini Robot.** The robot uses the Rapidly-exploring Random Tree (RRT) algorithm to locate disassembled components and transport them to the destination. The resulting path is executed via differential drive control, enabling smooth turns with the AGV robot. During delivery, the planned route is constrained to straight-line motion to enhance transportation reliability and ensure accurate placement at the assembly location.

**Humanoid Carrying Skills.** We formulate physics-based humanoid control as a goal-conditioned reinforcement learning problem and adopt the AMP-based single-object manipulation paradigm from previous research Peng et al. (2021); Gao et al. (2024). Style rewards encourage rapid postural dynamics such as quick recovery and linear locomotion, while target rewards guide precise object manipulation, enabling the humanoid to learn efficient carrying behaviors.

## 5 EXPERIMENTS

In this section, we present a comprehensive evaluation of **CLiMRS** to address the following questions:

(1) Is **CLiMRS** effective for simple daily-life multi-robot collaboration?

(2) Can **CLiMRS** perform well in challenging industrial scenarios with multi-robot assembly tasks?

(3) Through ablation studies, how critical are the individual components of **CLiMRS**?

Table 2: **Comparison Across Task Types in the COHERENT Benchmark. CLiMRS** outperforms all the baselines, achieving the largest gain on the most challenging trio-type tasks.

| Method | Mono-type Task | | Dual-type Task | | Trio-type Task | | Average | |
|---|---|---|---|---|---|---|---|---|
| | SR | AS | SR | AS | SR | AS | SR | AS |
| DMRS-1D | 0.700 | 10.6 | 0.467 | 18.0 | 0.667 | 20.7 | 0.600 | 17.2 |
| DMRS-2D | 0.500 | 11.5 | 0.267 | 19.9 | 0.400 | 24.5 | 0.375 | 19.6 |
| CMRS | **0.900** | 7.9 | 0.533 | 16.4 | 0.533 | 22.2 | 0.625 | 16.5 |
| Primitive MCTS | 0.000 | 14.0 | 0.000 | 21.5 | 0.000 | 26.9 | 0.000 | 21.7 |
| LLM-MCTS | 0.700 | 10.2 | 0.067 | 20.9 | 0.000 | 26.9 | 0.200 | 20.5 |
| COHERENT | **0.900** | 7.4 | **1.000** | 11.9 | **1.000** | 16.1 | **0.975** | 12.4 |
| **CLiMRS**(Ours) | **0.900** | **6.8** | **1.000** | **11.5** | **1.000** | **13.1** | **0.975** | **10.9** |
| Ground Truth (GT) | – | 6.5 | – | 10.3 | – | 12.9 | – | 10.3 |

Table 3: **Comparison Across Scenes in the COHERENT Benchmark. CLiMRS** outperforms all the baselines in every scene, demonstrating its superior performance.

| Method | S1 | | S2 | | S3 | | S4 | | S5 | | Average | |
|---|---|---|---|---|---|---|---|---|---|---|---|---|
| | SR | AS | SR | AS | SR | AS | SR | AS | SR | AS | SR | AS |
| DMRS-1D | 0.500 | 17.4 | 0.625 | 15.8 | 0.625 | 18.3 | 0.750 | 15.1 | 0.500 | 19.3 | 0.600 | 17.2 |
| DMRS-2D | 0.500 | 18.9 | 0.500 | 18.3 | 0.375 | 20.6 | 0.250 | 18.9 | 0.250 | 21.1 | 0.375 | 19.6 |
| CMRS | 0.875 | 13.1 | 0.625 | 16.6 | 0.625 | 18.5 | 0.375 | 18.1 | 0.625 | 15.9 | 0.625 | 16.5 |
| Primitive MCTS | 0.000 | 21.5 | 0.000 | 21.8 | 0.000 | 22.5 | 0.000 | 20.5 | 0.000 | 22.0 | 0.000 | 21.7 |
| LLM-MCTS | 0.250 | 20.0 | 0.250 | 20.4 | 0.250 | 21.3 | 0.125 | 19.9 | 0.125 | 20.9 | 0.200 | 20.5 |
| COHERENT | **1.000** | 13.1 | **1.000** | 11.4 | **1.000** | 11.9 | **1.000** | 11.4 | **0.875** | 14.0 | **0.975** | 12.4 |
| **CLiMRS**(Ours) | **1.000** | **10.8** | **1.000** | **10.4** | **1.000** | **11.8** | **1.000** | **10.4** | **0.875** | **11.4** | **0.975** | **10.9** |
| Ground Truth (GT) | – | 10.3 | – | 10.4 | – | 10.8 | – | 9.8 | – | 10.5 | – | 10.3 |

We evaluate **CLiMRS** in two distinct environments: **CLiMBench** and a simpler heterogeneous multi-robot collaboration benchmark from COHERENT Liu et al. (2025). For LLM api use, we use *gpt-4-0125-preview* to align with the setting in COHERENT. For quantitative analysis, we use task Success Rate (SR) and Average Step (AS) as evaluation metrics in this paper.

## 5.1 EVALUATING **CLiMRS** ON SIMPLE DAILY-LIFE MULTI-ROBOT COLLABORATION

To answer Question (1), we evaluate **CLiMRS** on the COHERENT benchmark, a simpler heterogeneous multi-robot benchmark that includes diverse tasks across five real-world scenes, but involves at most three heterogeneous robots and assumes perfect skill execution. We adopt its evaluation metrics and use the reported results as our baseline.

Results shown in Table 2 and 3 suggest that **CLiMRS** succeeds on nearly all COHERENT tasks and achieves higher efficiency with fewer steps. This trend holds across every scene, demonstrating our **CLiMRS**' superior performance. Notably, in the most challenging trio-type tasks, which require all three robots to collaborate, **CLiMRS** delivers the largest gain, reducing the Average Step count by 18.6%, indicating that our approach offers stronger improvements on more complex tasks.

## 5.2 EVALUATING **CLiMRS** ON **CLiMBENCH** WITH ROBOT ASSEMBLY TASKS

To answer Question (2), we evaluate **CLiMRS** on **CLiMBench**. Our baselines include the following:

- DMRS-1D: a variant of CoELA Zhang et al. (2024b), this decentralized framework lets robots determine their next step through dialogue, with the final decision summarized by the last robot.
- CMRS: a primitive centralized system Huang et al. (2022) that uses a single decision-making LLM to output executable actions, where all information is stored in the prompt.
- COHERENT: an approximately centralized approach combining an oracle planner LLM and feedback LLM for robots, where dialogue is passed through memory, forming a Proposal–Execution–Feedback–Adjustment cycle.

Table 4: **Comparison Across Tasks in CLiMBench. CLiMRS** outperforms all our baselines and reduces the Average Step (AS) by over 40%.

| Method | Task 1 (Easy) | | Task 2 (Easy) | | Task 3 (Hard) | | Task 4 (Hard) | | Average | |
|--------|------|------|------|------|------|------|------|------|------|------|
| | SR | AS | SR | AS | SR | AS | SR | AS | SR | AS |
| DMRS-1D | 0.000 | 15.0 | 0.000 | 15.0 | 0.000 | 19.0 | 0.000 | 19.0 | 0.000 | 17.0 |
| CMRS | 0.000 | 15.0 | 0.000 | 15.0 | 0.000 | 19.0 | 0.000 | 19.0 | 0.000 | 17.0 |
| COHERENT | **1.000** | 13.6 | 0.800 | 13.6 | 0.400 | 18.2 | 0.600 | 17.8 | 0.700 | 15.8 |
| **CLiMRS** (Ours) | **1.000** | **8.2** | **1.000** | **8.4** | **1.000** | **9.4** | **1.000** | **9.2** | **1.000** | **8.8** |
| Ground Truth (GT) | – | 7.0 | – | 7.0 | – | 9.0 | – | 9.0 | – | 8.0 |

Table 5: **Ablation Studies.** Removing dialogue history, feedback information, or the grouping stage significantly reduces both Success Rate (SR) and Average Step (AS).

| Method | Task 1 (Easy) | | Task 2 (Easy) | | Task 3 (Hard) | | Task 4 (Hard) | | Average | |
|--------|------|------|------|------|------|------|------|------|------|------|
| | SR | AS | SR | AS | SR | AS | SR | AS | SR | AS |
| **CLiMRS** w/o history | 0.000 | 15.0 | 0.000 | 15.0 | 0.000 | 19.0 | 0.000 | 19.0 | 0.000 | 17.0 |
| **CLiMRS** w/o feedback | 0.200 | 14.8 | 0.200 | 14.8 | 0.200 | 18.8 | 0.200 | 18.8 | 0.200 | 16.8 |
| **CLiMRS** w/o grouping | 0.600 | 14.0 | 0.800 | 13.2 | 0.600 | 17.2 | 0.600 | 17.4 | 0.650 | 15.5 |
| **CLiMRS** (Ours) | **1.000** | **8.2** | **1.000** | **8.4** | **1.000** | **9.4** | **1.000** | **9.2** | **1.000** | **8.8** |
| Ground Truth (GT) | – | 7.0 | – | 7.0 | – | 9.0 | – | 9.0 | – | 8.0 |

For quantitative evaluation, we fixed the scene parameters and selected four representative scenarios, manually deriving minimal-step solutions as ground-truth references. A task is deemed successful only if completed within twice the ground-truth step count. Due to stochastic skill execution in **CLiMBench**, we run each task five times and report mean Success Rate (SR) and Average Step (AS).

Results in Table 4 show that **CLiMRS** achieves 100% success in **CLiMBench**, surpassing every baseline. It also reduces the Average Step (AS) by 44.30% compared with the best baseline, a substantial efficiency gain highlighting the strength of **CLiMRS** for long-horizon heterogeneous multi-robot collaboration. Moreover, comparing baseline performance in Tables 2 and 4 reveals that the assembly tasks in **CLiMBench** are more challenging than those in the COHERENT benchmark, demonstrating the value of **CLiMBench** as a tougher testbed for heterogeneous multi-robot systems.

### 5.3 ABLATION STUDIES ON **CLiMRS**

To answer Question (3), we assess the necessity of each component of **CLiMRS** through: (i) removing the dialogue history, (ii) removing the feedback information, and (iii) removing the grouping stage from the grouping–planning–feedback–execution cycle. We use the same evaluation tasks and metrics as in Section 5.2, and the results are reported in Table 5. The results show that removing any of these components lowers the task success rate and markedly increases the average steps, underscoring the crucial roles of dialogue history, feedback information, and the grouping stage in our method.

## 6 CONCLUSION

In this paper, we present **CLiMRS**, a human-team-inspired adaptive negotiation paradigm for heterogeneous multi-robot systems. To evaluate these capabilities, we introduce **CLiMBench**, a heterogeneous multi-robot benchmark of challenging assembly tasks. Extensive experiments suggest that **CLiMRS** surpasses all baselines, boosting success rates and improving efficiency by over 40% on more complex tasks. Our results demonstrate that leveraging human-inspired group formation and negotiation principles markedly enhances the efficiency of heterogeneous multi-robot collaboration.

**Discussion and Limitation.** In this paper, we primarily aim to enhance the efficiency of multi-robot collaboration, while leaving inference latency and computational cost of the LLMs outside the present scope. Managing API costs under inference-efficiency constraints and exploring asynchronous inference–execution are promising aspects that we plan to investigate in future work.

## ETHICS STATEMENT

Our study investigates multi-robot collaboration using large language models (LLMs) for planning and negotiation. Below we address the main ethical considerations relevant to this work:

- **No Human or Sensitive Data.** Our research involves only simulated robotic environments and does not include human subjects, personally identifiable information, or sensitive real-world data.
- **Safety and Deployment.** Although our benchmark and methods are evaluated only in simulation, real-world deployment of autonomous multi-robot systems may present physical-safety risks. Any future use outside simulation should therefore incorporate rigorous testing, appropriate safety protocols, and adherence to all relevant regulations.
- **Potential Bias and Fairness.** The LLMs used are pretrained by third parties and may inherit societal biases. Our work does not amplify these biases in deployment scenarios; nevertheless, we acknowledge this limitation and recommend further bias auditing for any real-world applications.

The authors affirm compliance with the ICLR Code of Ethics and accept full responsibility for the integrity and societal implications of the research.

## REPRODUCIBILITY STATEMENT

We have taken extensive measures to ensure the reproducibility of our results. A full description of the **CLiMRS** framework is provided in Sec. 3. Details on the **CLiMBench** are provided both in Sec. 4 and in the Appendix. Evaluation protocols are reported in Sec. 5. An anonymous link to the source code and the Appendix is included in the supplementary material, allowing reproduction.

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
