# OpenReview forum: "CLiMRS: Cooperative Large-Language-Model-Driven Heterogeneous Multi-Robot System"
_ICLR.cc/2026/Conference — ICLR 2026 Conference Withdrawn Submission_

### Official Review · Reviewer_62Hf · 2025-10-28

**Soundness:** 2
**Presentation:** 2
**Contribution:** 2
**Rating:** 2
**Confidence:** 4

**Summary:**

This work presents CLiMRS, a framework for leveraging large language models (LLMs) for heterogeneous multi-robot collaboration tasks. A  major distinction between this and prior method in the same domain is using an adaptive negotiation framework, where different robots are dynamically formed into subgroups during task execution, and the grouping can be changed throughout a task.

For evaluation, the authors use both a simpler prior benchmark and introduce a new multi-robot benchmark, CLiMBench, which contains assembly tasks with multiple robot types and skill libraries. Empirically, CLiMRS significantly outperforms baseline methods and enhances the efficiency of heterogeneous multi-robot collaboration especially on complex tasks.

**Strengths:**

1. The proposed dynamic grouping formulation offers several advantages in multi-robot coordination, i.e. reducing communication overhead while enabling concurrent discussions, plan refinement, and parallel action execution.

2. The CLiMBench task environments show good simulation support for various robot types and skill primitives.

3. Experiment results show clear performance over basement methods.

**Weaknesses:**

1. Lack of qualitative results. Visualizations for the CLiMBench tasks are limited in both the main paper and the appendix pdf file. The empirical results would have been much more convincing if the authors showed task videos of the robots completing the tasks with the reported success rates.

2. Lack of considerations for real world robot deployments. Although the authors dedicated efforts into building diverse simulation tasks environments, it's difficult to access the applicability of the proposed framework to real world robot systems without real world results or a extensive discussion section on additional challenges for real world transfer. The authors claim the simulation tasks to represent 'real-world scenarios', but beyond the basic task definition, the simulation tasks do not properly reflect real world deployment requirements. A major challenge in real world would have been occlusion, i.e. as the AGV robots moving in a task space big as the proposed CLiMBench scenarios, the robots would not have been able to easily get the location and statuses of each other as in simulation where oracle simulation state information can be easily acquired.

3. Presentation quality. Sections 4 and 5 randomly switches between high-level task descriptions and low-level implementation details, which obstructs the readability of the manuscript. For example, in Line 358, mentioning the Franka robot task space controller would have been sufficient for the main text, but the authors added "uses the task-space inertia matrix and gravity compensation to compute joint torques, yielding a spring–damper response".

**Questions:**

1. In Table 1, what is this Franka skill really doing and how is it implemented in simulation? "[check] <franka>check <right wheel>"

2. How is RRT path planning done with multiple AGV robots moving at once? Do all the mobile cars move together?

3. What exactly is in the observation space for the Franka and AGV robots? The paper only mentioned observation parameters for the humanoid.

---

> ### Author Response · Authors · 2025-11-19
> **Official Comment: Response to Reviewer 62Hf**
>
> We sincerely thank the reviewer for the constructive comments and for **recognizing the benefits of our dynamic subgrouping formulation, the capability of CLIMBench, and the strong empirical performance of CLiMRS**. We address all concerns below and will incorporate the corresponding revisions into the updated manuscript.
>
> **(1) Qualitative results and visualizations**
>
> We agree that additional qualitative evidence would improve clarity. The revised version will include:
> (a) more visualizations of CLIMBench environments,
> (b) representative **execution rollouts**, and
> (c) an anonymized link to **video demonstrations**.
> These additions will make the reported behaviors more transparent.
>
> **(2) Real-world deployment considerations and occlusion**
>
> CLIMBench is designed as **a planning-focused benchmark**, where simulator state abstracts standard localization/communication modules (following common practice in multi-robot planning). Our focus is on heterogeneous coordination and robustness to execution uncertainty.
>
> Importantly, CLiMRS **naturally supports object-level occlusion at the task level**: in scenarios where relevant objects are not visible initially, the planner explicitly triggers a search-and-locate phase—e.g., dispatching an AGV to find the part, return an updated pose, and only then initiating manipulation. This mechanism allows CLiMRS to operate even when key objects start out-of-view.
>
> We fully acknowledge that perception-level occlusion in real robotic systems requires explicit sensing modules (cameras, LiDAR, on-board localization). In the revised manuscript, we will add a discussion clarifying this distinction and outlining how CLiMRS could be integrated with standard perception and sensing pipelines in future work. **Additional qualitative visualizations will illustrate the planning behavior in scenarios where objects start out-of-view in the simulated environment**.
>
>
> **(3) Presentation quality**
>
> Thank you for the suggestion. We will revise Sections 4 and 5 to keep high-level task descriptions and low-level implementation details clearly separated, removing unnecessary controller-level elaboration.
>
> **(4) Clarification of the Franka “check” skill**
>
> The “*[check]*” skill is a simple inspection primitive: the end-effector moves to a predefined pose to verify object readiness (e.g., location or accessibility) before grasping. In simulation, this is implemented via rule-based waypoint motion. We will clarify this in the manuscript.
>
> **(5) RRT path planning with multiple AGVs**
>
> AGVs use standard RRT planning:
> (a) **Static obstacles (walls, fixed structures, Franka bases) are included directly in the collision geometry**.
> (b) **Dynamic obstacles (e.g., boxes being manipulated by other robots) are handled naturally via subgroup coordination and execution dependencies**.
> (c) All AGVs move concurrently. A lightweight priority rule resolves rare conflicts or near-collisions, and AGVs replan locally when needed.
> We will expand the motion-planning description accordingly.
>
> **(6) Observation spaces for Franka and AGV**
>
> Thank you for pointing out the need for clearer observation definitions. We clarify below and will include the complete specifications in the revised manuscript. Because the Franka and AGV controllers rely on predefined, non-learning motion primitives, their observation requirements are much simpler than those of the humanoid. During task execution:
> (a) Franka: **Franka receives only the symbolic and geometric information necessary for performing assembly primitives**, including (i) the states of all parts (pose and type), (ii) the target placement pose for the current subgoal, and (iii) its own joint states during PD tracking. No additional perception or global state is used; this abstraction is standard for planning-focused benchmarks where manipulation controllers are non-learning and rely on structured task descriptions.
> (b) AGV. Each **AGV receives the minimal map information required for RRT-based navigation**, including (i) the static workspace layout (map boundary and fixed obstacles), (ii) the AGV’s own 2D pose, and (iii) the poses of other agents for collision avoidance. This corresponds to the typical map abstraction used in multi-robot navigation benchmarks. **No privileged simulator internals are accessed; all baselines operate under the same observation model**.
> (c) Humanoid. The humanoid observation space remains unchanged from the submission and will be summarized together with the above for completeness.
> We will incorporate these definitions and unify the observation-space description across all robot types in the revised manuscript.
>
>
> All methods, including baselines, operate under the same observation assumptions.
>
> We thank the reviewer again. With the added qualitative results, clearer motion-planning and observation descriptions, and an improved discussion of real-world considerations, we believe the revised manuscript will be substantially strengthened.

---

### Official Review · Reviewer_Jq68 · 2025-10-31

**Soundness:** 3
**Presentation:** 3
**Contribution:** 2
**Rating:** 4
**Confidence:** 4

**Summary:**

This paper introduces CLiMRS, a LLM–driven framework for heterogeneous multi-robot collaboration. Each robot is paired with an LLM agent, and the system organizes robots into dynamic subgroups for joint planning via a grouping–planning–feedback–execution loop inspired by human teamwork.

The authors also propose CLiMBench, a new benchmark built in IsaacGym for evaluating heterogeneous robot collaboration in realistic physics-based settings. Experiments show that CLiMRS improves task success and efficiency by over 40% compared to baselines such as COHERENT and CMRS, and achieves 100% success in benchmark tasks.

**Strengths:**

Novel Framework: Innovative use of dynamic subgrouping and negotiation principles inspired by human teamwork.
Benchmark Contribution: CLiMBench provides a realistic and valuable testbed for heterogeneous multi-robot systems.
Strong Experimental Results: Clear, consistent quantitative improvements across tasks; ablations validate each module’s contribution.
Clarity and Presentation: Well-organized writing, clear figures.

**Weaknesses:**

I am worried about novelty. Many previous works have discussed about the framework of LLM for multiple robot collaboration, and also the heterogeneous condition. The centralized planner + local planning and feedback have been proposed by many studies before.

The studied multiple robot condition and task is not difficult in the task planning level. The robot number is not large. Whether this framework can be scalable to robot number over 10 or 20 is not clear.

How to do the motion planning is not explained, by pre-defined rules? Then how to handle the cases with different motion planning difficulty such as the obstacles and objects.

**Questions:**

The current appendix is not in the same file as main article.

Robots handle different tasks with different execution time. Step by step planning and sync may not be effective enough. How to consider the case with asynchronous communication?

---

> ### Author Response · Authors · 2025-11-19
> **Official Comment: Response to Reviewer Jq68**
>
> We sincerely thank the reviewer for the positive and encouraging feedback, especially **regarding the novelty of our dynamic subgrouping mechanism, the value of CLIMBench, and the strength of our empirical results and ablations**. We address the reviewer’s concerns and questions below and will incorporate the corresponding clarifications into the revised manuscript.
>
> **(1) On novelty relative to prior centralized–local LLM planning frameworks**
>
> We appreciate the reviewer’s point regarding prior frameworks that combine a centralized planner with local planning and feedback. **Our contribution differs in that CLiMRS introduces adaptive, dynamically formed sub-groups that reorganize during execution based on spatial proximity, subgoal dependencies, and evolving task context**.
> While baselines use a fixed centralized–local structure, our subgrouping mechanism:
> (a) reduces context load for high-level planning,
> (b) enables parallel planning across multiple sub-groups, and
> (c) improves robustness when heterogeneous agents have different capabilities or skill dependencies.
>
> **(2) On scalability to >10 or >20 robots**
>
> We agree that large-scale multi-robot scenarios are important. In our current benchmark, scalability is constrained primarily by simulation environment complexity and the cost of constructing high-quality IsaacGym assembly tasks, rather than by limitations of our method.
>
> That said, **the design of CLiMRS inherently supports scalability**:
> (a) Dynamic subgrouping strategically limits global context sharing, so planning complexity grows approximately linearly, not quadratically, with team size.
> (b) Sub-groups operate in parallel, and only local interactions must be reasoned about.
> (c) Our **baseline methods already show reduced efficiency with only moderate numbers of agents**, while CLiMRS maintains strong performance, **demonstrating the advantage of our structured grouping**.
> We will add a discussion on scalability assumptions and future extensions using larger teams and additional benchmarks.
>
> **(3) On motion planning and handling complex environmental conditions**
>
> Thank you for raising this point. We clarify the motion planning pipeline below and will expand Section 4/5 and the appendix accordingly:
>
> (a) Humanoid skills are obtained via AMP-trained behaviors.
>
> (b) Franka and AGV skills are rule-based motion primitives integrated with standard planning modules (e.g., RRT for AGVs, task-space controllers for manipulators).
>
> (c) Static obstacles are explicitly incorporated into AGV RRT planning, ensuring collision-free paths.
>
> (d) Dynamic obstacles (e.g., boxes being moved by humanoids) are naturally handled through task dependencies and subgrouping: if a humanoid does not clear the object in time, the AGV cannot proceed, which becomes a test of team coordination rather than purely geometric planning.
>
> (e) Collision handling and priority rules help maintain safety—humanoids have highest priority, followed by Frankas, then AGVs, and agents perform local avoidance when needed.
>
> We will revise the manuscript to make these implementation details more explicit.
>
> **(4) On asynchronous communication and heterogeneous execution times**
>
> We agree with the reviewer that robots often require different execution durations. In our current evaluation, we use a synchronized cycle for consistency across baselines: each agent maintains a “task completion flag”, and we advance to the next LLM planning cycle when all sub-group actions for that cycle have completed.
>
> However, the CLiMRS architecture naturally supports asynchronous operation, as it is built around dynamic subgrouping:
> (a) **Sub-groups operate independently**, beginning execution as soon as local planning is completed.
> (b) Feedback-driven replanning allows **early-finished robots or delayed robots to trigger re-grouping or task reassignment without halting other agents**.
> (c) **Local planners run in parallel**, enabling different sub-groups to progress asynchronously.
> We will add a discussion of asynchronous extensions in the revised version.
>
> **(5)  Appendix formatting**
>
> Thank you for noting that the appendix PDF was not merged with the main paper. We will fix this in the revised submission.
>
> We appreciate the reviewer’s constructive feedback and openness toward acceptance. We believe the above clarifications and revisions will significantly strengthen the clarity and completeness of the manuscript.

---

### Official Review · Reviewer_m3z7 · 2025-11-01

**Soundness:** 2
**Presentation:** 3
**Contribution:** 2
**Rating:** 2
**Confidence:** 4

**Summary:**

This paper presents CLIMRS (Cooperative Large-Language-Model-Driven Heterogeneous Multi-Robot System), a novel framework for multi-robot collaboration designed to manage complex, long-horizon tasks for heterogeneous teams under uncertainty. The core of the proposed architecture is a "grouping-planning-feedback-execution" loop, which is inspired by human teamwork principles. In this system, a high-level "General Proposal Planner" dynamically forms agent sub-groups, which are then managed by parallel "Local Oracle Planners" that generate specific commands.

To evaluate this system, the authors also introduce CLIMBench, a new simulation benchmark built in IsaacGym. This benchmark is designed to test multi-agent assembly tasks and, notably, models realistic physics and skill execution failures, which the authors contrast against other benchmarks that assume perfect skill execution. The experimental results show that CLIMRS achieves a 100% success rate on CLIMBench, significantly outperforming baselines like COHERENT (70%) and other decentralized and centralized methods (0%).

**Strengths:**

1. The paper addresses a significant and challenging problem at the intersection of large language models and multi-robot systems, specifically focusing on heterogeneous agent collaboration. The core architectural idea of a human-inspired, dynamic grouping and negotiation framework  is a novel contribution to the field of multi-agent LLM-based planning.

2. A significant strength is the introduction of the CLIMBench benchmark. The authors correctly identify a limitation in existing work, such as the COHERENT benchmark , which assumes perfect skill execution. By building CLIMBench in a realistic physics simulator (IsaacGym) and explicitly modeling execution failures, the authors contribute a potentially valuable testbed for evaluating the robustness of planning frameworks, a critical aspect of real-world robotics.

3. The proposed CLIMRS framework is presented clearly, with its modular components and the information flow of the "grouping-planning-feedback-execution" loop well-articulated  (e.g., in Figure 2).

**Weaknesses:**

1. The paper's primary claim of a ">40% efficiency" improvement  rests on an insufficient metric. This gain is measured only by the "Average Step (AS)" count. This metric fails to capture the computational cost and, more importantly, the inference latency of the system. The proposed framework involves a sequential hierarchy of multiple LLM calls (General Planner, Local Planners, Agent Executors) within each loop. In any practical robotic application, wall-clock time is a dominant, if not the most critical, measure of efficiency. As the authors acknowledge in their limitations section, "inference latency and computational cost of the LLMs [are] outside the present scope". Without an analysis of wall-clock time, the claim of improved efficiency is unsubstantiated, as a lower step count could easily be offset by significantly longer per-step planning time.

2. A significant ambiguity in the methodology lies in the "Local Oracle Planner". In robotics and machine learning, the term "oracle" typically implies access to privileged, ground-truth information (e.g., perfect global state, simulator-internal data) not available to a deployed agent. The paper never defines what "oracle" signifies in this context. If these planners possess such privileged information, it would (a) contradict the paper's positioning of operating under "spatial constraints and execution uncertainties"  and (b) constitute an unfair advantage over the baseline methods, which would compromise the validity of the experimental comparison.

3. The feedback mechanism described in Section 3.3 appears to rely on unrealistically strong assumptions. The "Agent Executor" LLM is tasked with categorizing failures at a high level of causal abstraction, such as "(1) improper grouping" or "(2) incorrect agent selection". It is unclear how a local agent executor, whose scope is to verify and execute a specific command, could possess the global context necessary to deduce such a high-level, systemic reason for failure. This feedback provides an information-rich, structured signal that directly guides the high-level planner. If the baseline methods do not have access to a comparable feedback signal, this represents a significant confounding variable that, much like a potential oracle, could explain the performance gap.

4. The authors propose both the method and the primary benchmark (CLIMBench) used for evaluation. The results on CLIMBench (Table 4) show a stark disparity: CLIMRS achieves a 100% success rate, while the DMRS and CMRS baselines score 0%. This result suggests the benchmark's challenges may be specifically tailored to the dynamic grouping architecture of CLIMRS. On this benchmark, the performance gap is negligible: CLIMRS and the COHERENT baseline both achieve an identical 97.5% average success rate. The fact that CLIMRS’s overwhelming superiority vanishes on a benchmark not designed by the authors is a questionable observation.

**Questions:**

1. Could the authors provide wall-clock time comparisons for task completion, in addition to "Average Step"? This would account for the cumulative LLM inference latency of the CLIMRS framework versus the baselines

2. Can the authors please provide a precise definition of the "Local Oracle Planner"?

3. How does a local "Agent Executor" LLM obtain the necessary global context to provide high-level causal feedback like "improper grouping"? Is this information provided to the agent, or is it expected to infer it? How does this compare to the feedback available to the baseline methods?

4. How do the authors explain the performance discrepancy between CLIMBench and the COHERENT benchmark? Why do the DMRS and CMRS baselines, which are established architectures, fail with 0% success on CLIMBench while performing reasonably on COHERENT?

---

> ### Author Response · Authors · 2025-11-19
> **Official Comment: Response to Reviewer m3z7**
>
> We sincerely thank Reviewer m3z7 for the detailed and thoughtful feedback. We appreciate Reviewer m3z7’s recognition of the **novelty of our dynamic grouping framework and the value of CLIMBench as a realistic heterogeneous multi-robot benchmark**. We address all concerns point-by-point below and will incorporate the corresponding clarifications into the revised manuscript.
>
> **(1) On the use of Average Step (AS) and wall-clock efficiency**
>
> We fully agree that wall-clock time is an important practical measure of efficiency. In our system, however, **the duration of each planning–execution cycle is dominated by robot execution time, while LLM inference contributes only a fraction of the total latency**. Moreover, CLiMRS introduces at most one additional LLM call per cycle relative to the strongest baseline (COHERENT).
> As a result, the differences in LLM inference latency across methods are negligible relative to execution time, making **AS a reliable proxy for overall planning–execution efficiency in our setting**.
>
> In addition, we note that **the LLM latency observed in our experiments is largely due to using public API endpoints**. In realistic industrial deployments, LLMs are typically deployed on local servers or edge devices, which substantially reduces inference latency and further diminishes the relative impact of per-step LLM calls across different methods.
>
> We have begun running wall-clock measurements under identical hardware and API conditions. We will include the full results in the revised version.
>
> **(2) Definition and role of the “Local Oracle Planner”**
>
> We appreciate the reviewer’s comment and acknowledge that the term “oracle” is misleading. The reviewer is correct that **this module is more accurately described as a *sub-group manager***, and we will update the terminology accordingly.
>
> Crucially, **this component does NOT access any privileged simulator-internal information**. It operates solely on:
> (a) the local observations of the robots in its sub-group, and
> (b) the execution-history-based feedback maintained in context.
>
> Thus, the module neither contradicts the setting of “spatial constraints and execution uncertainties,” nor provides any unfair advantage over baselines. We will clarify its definition, inputs, and decision process in the revised paper.
>
> **(3) On the feedback mechanism and the availability of global state**
>
> We agree that our original description may have caused confusion. We clarify that:
> (a) Agent Executors do not observe global state, global spatial configuration, or cross-group information.
> (b) **The failure categories** mentioned in the paper (e.g., “improper grouping”) are not produced by the executors. They **arise from our post-hoc analysis of observed failure patterns**.
> (c) At runtime, executors output only local, plan-conditioned signals (e.g., inability to complete a skill or local inconsistencies).
> (d) **The top-level planner interprets these signals inductively over execution history**, not through access to explicit global failure labels.
>
> Thus, the system performs adaptation based on local feedback aggregation, not on privileged global diagnostics. We will add a clearer description in the revised manuscript.
>
> **(4) On the CLIMBench vs. COHERENT benchmark performance discrepancy**
>
> We clarify that the observed gap results from differences in task complexity and step-efficiency requirements, not from benchmark bias. Both CLIMBench and COHERENT **define success using a step threshold**; however, CLIMBench tasks involve richer heterogeneous dependencies, multi-stage sequencing, and realistically modeled execution failures. In these more demanding settings, baseline methods—especially CMRS and DMRS—produce very long execution horizons and frequently exceed the step bound, **leading to 0% success under the time constraint**.
>
> In addition, CLIMBench includes tasks of different difficulty levels, each with different optimal (ground-truth) step counts due to their structural complexity. We observe that the strong baseline method (COHERENT) exhibits different success rates across these difficulty tiers, which further reflects its sensitivity to step efficiency under increasing coordination complexity.
>
> In contrast, the tasks in COHERENT are considerably simpler (except one that is inherently unsolvable under the original setup), allowing baselines to complete tasks within the threshold. Moreover, **COHERENT assumes full observability of the scene and deterministic skill execution**—once an LLM outputs a subgoal, it is treated as guaranteed to succeed. These assumptions reduce the uncertainty and execution variability encountered in CLIMBench, and naturally lead to higher success rates for baseline methods on COHERENT.
>
> We appreciate the reviewer’s constructive suggestions, which have helped us significantly improve the clarity of the paper. **All of the above clarifications will be incorporated into the revised manuscript**.

---

### Author Response · Authors · 2025-11-19
**Official Comment: General Summary (Reviewer-Specific Responses Submitted)**

We sincerely thank all reviewers for their thoughtful and constructive feedback. **We are encouraged that all reviewers highlighted important strengths of our work**, including

**(1) the novelty of our dynamic subgrouping framework inspired by human teamwork**,

**(2) the contribution of CLIMBench as a realistic and heterogeneous multi-robot benchmark**, and

**(3) the strong empirical performance, clear system modularity, and ablation results that validate each component**.

Across the three reviews, **the major concerns primarily relate to clarity**, particularly regarding (a) the interpretation of efficiency metrics, (b) the terminology and role of the “Local Oracle Planner,” (c) the scope of the Agent Executor’s feedback, and (d) the performance gap between CLIMBench and the COHERENT benchmark. We are grateful for these comments, which helped us identify several places where the exposition can be improved.

At a high level, we clarify that:

(1) **Average Step (AS) is a reliable proxy for wall-clock efficiency in our setting**, as cycle duration is dominated by robot execution time rather than LLM latency. We have begun running timing measurements under identical hardware and API conditions, and full measurements will be included in the revised manuscript.

(2) **The component referred to as the “Local Oracle Planner” does not access privileged information**; a more accurate name is ***sub-group manager***. It operates solely on local observations and within-group feedback, and we will update terminology and descriptions accordingly.

(3) **Agent Executors do not observe global state and do not classify failures**. The failure categories shown in the paper (e.g., “improper grouping”) come from our post-hoc analysis rather than executor outputs. At runtime, the system relies only on local, plan-conditioned feedback and contextual reasoning over execution history.

(4) **The performance gap between CLIMBench and COHERENT arises from differences in task complexity and step-efficiency requirements**. CLIMBench tasks involve richer coordination structure and multi-stage dependencies, causing baselines to exceed step thresholds even when partial progress is made. We will add a comparative analysis to clarify this distinction.

We will incorporate all these clarifications—along with improved explanations, terminology updates, and additional timing results—into the revised version, which is currently being prepared.
Finally, we note that **we have provided detailed reviewer-specific official comments addressing every question and concern raised by each reviewer**. We hope that these clarifications help resolve the ambiguities identified and strengthen the overall presentation of our work.

---

### Note · Authors · 2025-12-28

I have read and agree with the venue's withdrawal policy on behalf of myself and my co-authors.